# Development of Cellular Models to Study Efficiency and Safety of Gene Edition by Homologous Directed Recombination Using the CRISPR/Cas9 System

**DOI:** 10.3390/cells9061492

**Published:** 2020-06-18

**Authors:** Sabina Sánchez-Hernández, Araceli Aguilar-González, Beatriz Guijarro-Albaladejo, Noelia Maldonado-Pérez, Iris Ramos-Hernández, Marina Cortijo-Gutiérrez, Rosario María Sánchez Martín, Karim Benabdellah, Francisco Martin

**Affiliations:** 1Genomic Medicine Department, GENYO, Centre for Genomics and Oncological Research, Pfizer-University of Granada-Andalusian Regional Government, Parque Tecnológico Ciencias de la Salud, Av. de la Ilustración 114, 18016 Granada, Spain; sabina.sanchez@genyo.es (S.S.-H.); araceli.aguilar@genyo.es (A.A.-G.); beaguialb@correo.ugr.es (B.G.-A.); noelia.maldonado@genyo.es (N.M.-P.); iris.ramos@genyo.es (I.R.-H.); marina.cortijo@genyo.es (M.C.-G.); karim.benabdel@genyo.es (K.B.); 2Department of Medicinal & Organic Chemistry and Excellence Research Unit of “Chemistry Applied to Biomedicine and the Environment”, Faculty of Pharmacy, University of Granada, Campus de Cartuja s/n, 18071, Granada, Spain; rmsanchez@go.ugr.es

**Keywords:** models, DNA donor, on-target integration, homologous directed recombination (HDR), efficacy, safety, specificity, eGFP, dsRED, CRISPR/Cas9

## Abstract

In spite of the enormous potential of CRISPR/Cas in basic and applied science, the levels of undesired genomic modifications cells still remain mostly unknown and controversial. Nowadays, the efficiency and specificity of the cuts generated by CRISPR/Cas is the main concern. However, there are also other potential drawbacks when DNA donors are used for gene repair or gene knock-ins. These GE strategies should take into account not only the specificity of the nucleases, but also the fidelity of the DNA donor to carry out their function. The current methods to quantify the fidelity of DNA donor are costly and lack sensitivity to detect illegitimate DNA donor integrations. In this work, we have engineered two reporter cell lines (K562_SEWAS84 and K562GWP) that efficiently quantify both the on-target and the illegitimate DNA donor integrations in a *WAS*-locus targeting setting. K562_SEWAS84 cells allow the detection of both HDR-and HITI-based donor integration, while K562GWP cells only report HDR-based GE. To the best of our knowledge, these are the first reporter systems that allow the use of gRNAs targeting a relevant locus to measure efficacy and specificity of DNA donor-based GE strategies. By using these models, we have found that the specificity of HDR is independent of the delivery method and that the insertion of the target sequence into the DNA donor enhances efficiency but do not affect specificity. Finally, we have also shown that the higher the number of the target sites is, the higher the specificity and efficacy of GE will be.

## 1. Introduction

Recent advances in gene editing (GE)techniques, especially since CRISPR systems stormed into this field, are bringing precision medicine treatments closer to clinical implementation [1,2]. These techniques allow to modify the genome in a selective and precise way by using site-specific nucleases (SNs) such as Zinc-finger nucleases (ZFN) [3,4,5], transcription activator-like effector nucleases (TALEN) [6,7] and CRISPR systems [8,9,10]. These SNs are programmed to generate double-strand breaks (DSBs) in target DNA sequences that cells must repair. GE technologies make use of these cellular repair mechanisms to rewrite that particular DNA region, repairing an existing mutation, generating a new one or inserting a new DNA fragment [11]. DSBs can be repaired mainly by means of non-homologous end joining (NHEJ) or homology-directed repair (HDR). NHEJ does not require a DNA donor, so it generates small insertions or deletions (indels) that can be also used therapeutically [1,2,12,13,14]. On the other hand, HDR uses single- or double-stranded DNA templates with homology to the break site in order to either repair an existing mutation or to insert a new DNA fragment into a precise location [15,16].

HDR is currently considered the most accurate GE strategy and therefore, it is the strategy of choice for several groups. HDR has been studied for the development of gene therapy treatments for cancer [17,18] and several genetic diseases such as beta-haemoglobinopathies [19,20], primary immunodeficiencies (PID) [21,22], and metabolic storage disorders [23,24]. However, there are several limitations in the HDR strategies that preclude its clinical translation, including low efficiency in some relevant target cells, as HDR only occurs in late S/G2 phase [25,26,27,28,29]. In addition, DNA donor templates may be inserted in undesired sites generating an additional safety concern that sum up to the generation of off-target cleavage sites [27,28,29,30,31,32].

To overcome the limitations related to the nuclease activity, different groups have developed more efficient and specific CRISPR systems by including modifications in the sgRNAs [33,34,35] and different CRISPR/Cas variants [36]. Similarly, to overcome limitations that are specific for the HDR-GE strategies, studies have focused on multiple approaches such as the use of NHEJ inhibitors [37,38,39], the overexpression of factors involved in HDR [40], the control ofCas9 expression [41,42], the use of Cas9 chimeras harbouring different domains that stimulate HDR pathway [43,44] and of course, the optimisation of DNA donor templates. The nature of DNA donor (dsDNA versus ssDNA), the size of the donor (ssODNs versus plasmids) and the delivery system used to transfer DNA donor into target cells have been the most fundamental aspects to consider. An important aspect to consider for donor choice is the size and type of the desired genomic modification. For small modifications, up to 100 bp, ssODNs can be used efficiently. However, the insertion of a full length cDNA into a define gene or a complete expression cassette into a safe harbour locus, require larger DNA donors. We have centred our study on the development of cellular models to investigate the efficacy and specificity of large DNA donors. Plasmid donors have been commonly used and different approaches have been applied to improve HDR-GE efficiency [45,46]. Linearization of the plasmid [45] and the insertion of the target sequence recognised by the sgRNA outside of the homologous arms [47] increase HDR. Other GE strategies based on DNA donors have been developed such as the homology-independent targeted integration (HITI) or the precise integration into target chromosome (PITCH) [48,49,50,51]. Both strategies have shown that the inclusion of the nuclease target sequence into the DNA donor facilitates the insertion of the DNA donor by NHEJ (HITI) or microhomology-mediated end joining (MMEJ).

Although it has been less studied, illegitimate integration of DNA donor is another problem that needs to be understood and quantified before clinically apply HDR-GE strategies. As it has been previously mentioned, DNA donor templates may insert at off-target sites, either at off-target sites introduced by the SNs, at naturally occurring DNA breaks or at sites of microhomology. In general, the strategies aimed to determine the donor integration sites are based on the combination of PCR and sequencing techniques, being the linear amplification-mediated PCR(LAM-PCR) the most widespread [52,53]. In recent years, advances in genomic technologies have enabled a more accurate determination of donor integration in promiscuous sites [54,55,56]. However, dispersed illegitimate integration sites might not be detected using these approaches if the sequencing reads do not reach the minimum depth required [57]. This will lead to ignore the insertion of donors when occurring dispersed through the genome and not in specific sites [58]. Therefore, if the DNA donor integrates into naturally occurring breaks or into low-abundance microhomology sites, WGS will not detect them due to the low frequency of each integration event. This highlights the need to develop new systems that make it possible to detect and quantify the illegitimate integrations of DNA donors in a simple and cheap way. Different cell reporter systems have previously been described to analyse the repair pathway choice at DNA breaks by flow cytometry [59,60,61]. These models are based on the “Traffic light reporter”, where the GFP gene is linked with a mCherryFP using a T2A self-cleaving sequence. All these models include an SN target sequence on the reporter genes to turn off the expression of both fluorescence proteins. In another study, Olsen et al. developed a reporter 293-cell line containing a single copy of mutated eGFP gene (mGFP), where they observed a 3.5–3.6-fold increment of illegitimate integration levels when the donor included the ZFN-target sequence [29].The fact of using reporter genes as the target for SN in all these cellular models hinders the possibility of adapting these reporter cells to study the efficiency and safety of GE strategies, as the reporter genes are not relevant target for therapeutic interventions. Indeed, these models do not permit to evaluate off-target SN activity or quantify the levels of illegitimate donor integrations, which makes it uncommon to use these reporters for evaluating GE tools. 

Here, we have designed two reporter cellular models (K562 GF-WAS-P and K562 SEWAS84) which allow us to compare the efficiency and safety of CRISPR/Cas9 mediated HDR strategies based on large donor DNAs and directed to *WAS* sequences. In both models, differential fluorescence patterns define the efficacy and specificity of homologous directed recombination (HDR) in a trustworthy and unbiased way. On these models we have been able to evaluate the suitability of different delivery systems used to transfer the nuclease and large DNA donor templates to the target cells, as well as different donor configurations. By using these models, we have found that the specificity of HDR is independent of the delivery method and that the insertion of the target sequence into large DNA donor enhances efficiency but do not affect specificity. Finally, we also showed that the higher the number of the target sites is, the higher the specificity and efficacy of GE will be.

## 2. Material and Methods

### 2.1. Cell Lines and Culture Media

293T Cells (CRL11268; American Type Culture Collection; Rockville, MD, USA) were maintained in Dulbelcco’s Modified Eagle’s Medium (DMEM, Thermo Fisher Scientific, Waltham, MA, USA) with GlutaMAX^TM^ supplemented with 10% Foetal Bovine Serum (FBS, Biowest, Nuaillé, France) and 1% penicillin/streptomycin (Biowest) at 10% CO_2_ and 37 °C. The human cell line K562 (lymphoblast from bone marrow chronic myelogenous leukaemia, CCL-243) was obtained from ATCC (Manassas, Virginia, USA), and maintained in RPMI media (Biowest), supplemented with 10% FBS and 1% penicillin/streptomycin at 5% CO_2_ and 37 °C.

### 2.2. Lentiviral Constructions

The lentiviral plasmid SEWAS84 was obtained by an incorporation of a fragment WAS84 in the lentiviral plasmid SE [62]. The WAS84 fragment was generated by PCR of gDNA from K562 cells with the following primers; BamHI-WAS84 Fw (GGATCCATCCTCCCGCTCCTCCTTTCC) and BamHI-WAS84 Rv (GGATCCATCTTCCTGGGAAGGGTGGATT). The PCR product was purified using QIA quick PCR product purification kit (Qiagen, Hilden, Germany) and it was cloned into the pCR2.1-TOPO plasmid (pCR2.1 TA Cloning Kit, Thermo Fisher, Waltham, MA, USA) obtaining pCR2.1 WAS84 plasmid. Then, the SE and pCR2.1 WAS84 plasmids were digested with BamHI (New England Biolabs, Ipswich, MA, USA) and the resulting plasmid were ligated with T4 DNA ligase (New England Biolabs). After ligation and transformation into *E. coli* Stbl3-competent bacteria (Life Technologies, Thermo Fisher Scientific, Waltham, MA, USA), the plasmid was obtained using Wizard^®^ Plus SV Minipreps DNA Purification System (Promega, Madison, WI, USA). Restriction pattern was performed and the whole plasmid was eventually sequenced. Maxi-production was performed using NucleoBond^®^Xtra Maxi (Macherey-Nagel, Düren, Germany).

The lentiviral plasmid GWP was obtained after some subclonings. First, a fragment of 811 bp of the first intron of the *WAS* gene was generated by PCR of gDNA from K562 cells with the following primers hWASP2 Fw (AGGGTTCCAATCTGATGGCG) and hWASP2 Rv (TTGAGAACTGGCTTGCAAGTCC). As well as in the SEWAS84 LV plasmid, this PCR fragment was purified using the QIA quick PCR product purification kit (Qiagen, Hilden, Germany) and it was cloned into the pCR2.1-TOPO plasmid (pCR2.1 TA Cloning Kit, Thermo Fisher, Waltham, MA, USA), obtaining the plasmid pCR2.1 WAS811. Afterwards, a fragment of 387 bp of the first intron of *WAS* gene was amplified from the pCR2.1 WAS811 plasmid with the primers hWASP-I1Pfo Fw (TCCTGGACAGGACCACGAGAAC) and hWASP-I1Pfo Rv (TCCAGGACAGCGCCAGGTACAG), and this WAS387 fragment was cloned into the pCR2.1-TOPO plasmid, obtaining the plasmid pCR2.1 WAS I1 387. On the other hand, by means of PCR site-directed mutagenesis, the first ATG was removed from the codified eGFP sequence using the primers BamHI -δGFPFw (GGATCCTGAGCAAGGGCGA) and XhoI- GFP Rv (CCCTCGAGGTCGACTCTAGAGTC), from the SE plasmid. Again, this PCR product was cloned into the pCR2.1-TOPO plasmid, generating pCR2.1 GP plasmid. Then, the 387 bp fragment in pCR2.1 WAS I1 387 plasmid was cloned into pCR2.1 GP plasmid after digestion with PfoI (New England Biolabs, Ipswich, MA, USA), obtaining the plasmid pCR2.1 GWP. Finally, the pCR2.1 GWP and the SE plasmids were digested with BamHIand XhoI (New England Biolabs, Ipswich, MA, USA). The GWP fragment and the linearised SE were ligated with T4 DNA ligase (New England Biolabs, Ipswich, MA, USA) to obtain the GWP plasmid. After ligation and transformation into *E. coli* Stbl3-competent bacteria (Life Technologies, Thermo Fisher Scientific, Waltham, MA, USA), the plasmid was obtained using Wizard^®^ Plus SV Minipreps DNA Purification System (Promega, Madison, WI, USA). Restriction pattern was performed and the whole plasmid was eventually sequenced by Sanger. Maxi-production was performed using NucleoBond^®^Xtra Maxi (Macherey-Nagel, Düren, Germany).

### 2.3. Donor Constructions

The SdR*GFP plasmid, used as donor in the K562 SEWAS84 (eGFP-ON/dsRED-OFF) cell line, is formed by the SFFV (Spleen focus-forming virus) promoter, the dsRED cassette and the poliA signal from bovine Growth hormone (bGH), and the eGFP cassette with some different mutations in order to prevent its expression. This construction was synthesised by GenScript (Piscataway, NJ, USA) and was cloned in the puC57 plasmid. The SdR*GFPWAS84 plasmid was generated by digestion with PstI of the WAS84 fragment (from pCR2.1 WAS8 plasmid) and SdR*GFP plasmid and their subsequent ligation. 

The SEED lentiviral plasmid, used as DNA donor in HDR studies in the K562GWP (eGFP-OFF/dsRED-ON) cell line, was obtained cloning the cassette EF1α-dsRED in the SE plasmid downstream the GFP cassette in the same orientation. For this aim, EF1α-dsRED fragment was obtained from the pGEMT- EF1α-dsRED plasmid digested with KpnI (New England Biolabs, Ipswich, MA, USA) and it was cloned in the only KpnI site from the SE lentiviral backbone. SECdR and WAS_SECdR plasmids were generated by the inclusion of the CMV (cytomegalovirus) promoter and_dsRED_PoliA cassette in the SE, SE_WAS plasmids. Before this, the 387 bp fragment of the first intron of *WAS* gene was digested with EcoRI (New England Biolabs, Ipswich, MA, USA) from pCR2.1 WAS I1 387 plasmid in order to clone it before the SFFV promoter, generating the WAS_SE plasmid.

### 2.4. CRISPR/Cas9 Constructions

We used CRISPOR design tool (CRISPOR.org) to design our different gRNAs, whose sequences are gRNA1 (CGTCATAATCCACCCTTCCC) and gRNA9 (GAGGCAGGAAGGACCAGGTC). These two guides were synthesised by GenScript, together with the scaffold RNA and the U6 promoter, to ensure robust expression of the gRNA molecule (plasmid pU6_gRNA1/9). These fragments were cloned into SCas9WP plasmid [28] which expresses Cas9 under control of SFFV promoter. The plasmids pU6_gRNA1/9 and SCas9WP were digested with EcoRI (New England Biolabs, Ipswich, MA, USA), and subsequently ligated, resulting in the Lg1SCas9 and Lg9SCas9.

### 2.5. Vector Production and Virus Titration

Lentiviral vectors (LVs)were generated by transient co-transfection of 293T cells using (Lg1SCas9 and Lg9SCas9) vector plasmids with the human immunodeficiency virus (HIV) packaging plasmid pCMvdR8.91 (http://www.addgene.org/Didier_Trono) for integration-competent LVs or pCMvdR8.91D64V for IDLVs (http://www.addgene.org/; Plasmid #22036), and the plasmid pMD2.G.47 encoding the vesicular stomatitis virus (VSV-G) envelope gene (http://www.addgene.org/Didier_Trono). Briefly, 293T cells were planted on Petri dishes (Sarsted, Newton, NC, USA), in order to ensure 80% of confluence for transfection. The vectors, the packaging and envelope plasmid (in proportion 3-2-1) were resuspended in LipoD293 (SignaGen, Gaithersburg, MD, USA), as described previously [63]. The plasmid-LipoD293 mixture was added to pre-washed cells and incubated for five hours, when new medium was added. After 48 and 72 h, viral supernatants were collected, filtered through a 0.45µm (pore size) filter (Nalgene, Rochester, NY, USA) and concentrated either by ultrafiltration at 2000 g at 4 °C, using 100KD centrifugal filter devices (Amicon^®^ Ultra-15; Millipore, Billerica, MA, USA) or by ultracentrifugation (23,000 rpm, 2 h, at 4 °C; Beckman Coulter, Brea, CA, USA). The concentrated viruses were aliquoted, and storage at −80 °C.

Viral titres were determined by transducing 100,000 K562 cells in a 48-well tissue culture plates (BD Biosciences, San Jose, CA, USA). In the case of GFP-ON vectors, we use a 4-fold serial dilution of the supernatants and 5 days post-transduction, the percentage of eGFP-positive cells was assessed by fluorescence activated cell-sorting (FACS). For the rest of LVs, we obtained the genomic DNA kit QiAamp DNA Mini Kit (Qiagen, Hilden, Germany) after 5 days, and we carried out a qPCR in order to calculate the copy number per cell so that the titre could be calculated as previously described [63]. We used genomic DNA of 1 copy per cell to do a standard curve. The titres were calculated taking into account the number of cells plated, the volume of virus and the standard curve. We used KAPA SYBR FAST Universal qPCR (KAPA Biosystems, Wilmington, MA, USA)) in a Mx3005P QPCR System from Stratagene (Agilent Technologies, Santa Clara, CA, USA). The primers used for titration were ΔU3 (fw: GACGGTACAGGCCAGACAA) and PBS (rev: TGGTGCAAATGAGTTTTCCA).

### 2.6. Cell Transduction

We centrifuged at 1250 rpm for 5 min and counted. In total, 100,000 cells were plated in 48-well plates (BD Biosciences, San Jose, CA, USA) and incubated for 5 h with different multiplicity of infection (MOIs) of viral particles. After 5 h, the cells were washed in order to eliminate different rests of vector.

### 2.7. Copy Number (qPCR)

We used genomic DNA of 1 copy per cell to do a standard curve. This way, we could calculate the copy number. We used KAPA SYBR FAST Universal qPCR (KAPA Biosystems) in a programme (95 °C × 3″ + 60 °C × 30″) × 40 cycles + (95 °C × 1′ + 55 °C × 30″+ 95 °C × 30′) on a Strata gene MX3005P System (Agilent Technologies). The sequences of the primers were ΔU3 (fw: GACGGTACAGGCCAGACAA) and PBS (rev: TGGTGCAAATGAGTTTTCCA).

### 2.8. Single Cell Cloning

After sorting the different populations, we performed single cell clones. We centrifuged sorting cells to 1200 rpm for 5 min and washed cells with PBS. Then, we counted the cells and we plated 32 cells in a 96-wellplate and we add 200 μL of medium per well.

### 2.9. Cas9 RNP Assembly

To form RNP, the chemically synthesised crRNA and tracrRNA obtained from Synthego (Silicon Valley, CA, USA) (200 µM) were mixed and incubated following manufacturer’s instructions to form guide RNA (gRNA) at a concentration of 30 µM. Next, this gRNA was mixed in a 1:3 ratio in terms of volume with High fidelity Cas9 (IDT, Coralville, IA, USA) and incubated at room temperature 15 min to form RNP. Then, it was delivered to cells by means of nucleofection.

### 2.10. Nucleofection

K562 cells (2 × 10^5^) were nucleofected with 1 μg of the different donor plasmids (SdR*GFP, SdR*GFPWAS84, SEED, SECdR, or SECdR_WI1R) in two different ways: only the plasmids for the controls or together with the selected CRISPR system (1 μg of plasmids (Lg1 or Lg9) Lg1 or Lg9 IDLV (100 μL of concentrated supernatants added to cells immediately after nucleofection), or RNP). Nucleofection was performed with an Amaxa^®^ 4-D Nucleofector and a solution SF cell line (Lonza, Basel, Switzerland), applying FF-120 programme and following the nucleofection protocol for K-562 cells. The transfection efficiency was determined 48 h after nucleofection by flow cytometry.

### 2.11. Flow Cytometry

The expression pattern of eGFP and dsRED in control and nucleofected cells was determinedby flow cytometry. In all experiments, nucleofection of target cells with donor plasmid in the absence of nuclease was used as control. Cells of both models (K562_SEWAS84 and K562GWP) (1–4 × 10^5^) were collected and washed with cold FACS buffer (PBS containing 2% FBS and 0.5% bovine serum albumin (BSA)). Additionally, the expression of eGFP and dsRED was measured using a FACSCanto II flow cytometer (Becton Dickinson, Franklin Lakes, NJ, USA) with FACSDiva analysis software (BD Biosciences, San Jose, CA, USA). In all experiments, the first readout was measured 48 h after nucleofection, and consecutive read outs were carried out until day 30, when episomal expression of donor plasmids was completely extinguished. The relative efficiency of HDR was determined by means of the following formula:(1)$$\begin{array}{ll} HDR\;relative\;efficiency & =\frac{\%\;on-targetintegration\left(d30\right)\times 100}{\%Cells\;expressing\;donor\;plasmid\left(48\;h\right)}\\ & -\frac{\%\;remaining\;integrations\;in\;control\;cells\left(d30\right)\times 100}{\%\;Control\;cells\;expressing\;donor\;plasmid\left(48\;h\right)} \end{array}$$

### 2.12. Statistical Analyses

All data are represented as means ±SEM. Statistical analysis was performed using GraphPad Prism software (GraphPad Software, LaJolla, CA, USA; https://www.graphpad.com). We applied Mann–Whitney U test for the comparison of two independent samples (*p* < 0.05) and the Dunn’s test for multiple comparisons (*p* < 0.05).

## 3. Results

### 3.1. Generation of the eGFP-OFF/dsRED-ON Cellular Model

We first generated a reporter cell line in which on-target integrations due to HDR or to HITI could be detected in the same population. We transduced K562 cells (MOI = 0.1) with a LV that contains an 84 bp fragment of the *WAS* locus upstream of the eGFP (the SEWAS84 LV, see M&M for details). The integration of this LV into the genome of K562 cells make possible to use gRNAs targeting *WAS* sequences to edit the *eGFP* gene. eGFP+ cells were then sorted to generate the K562_SEWAS84 cell clone that harbours 1 vector/cell (Figure 1A). A drawing of the inserted vector genome on the reporter cell is depicted at the bottom of Figure 1A. We next designed a DNA donor (SdR*GFP) containing a dsREDcDNA flanked by a 5’ homologous arm (HA) harbouring the SFFV promoter and a 3’HA containing aneGFPcDNA with several mutations in it sequence. Therefore, when this donor is inserted in the targeted locus either by HDR or by HITI (see Appendix A), eGFP will be silenced and dsRED will be expressed (eGFP-OFF/dsRED-ON). However, if the donor is inserted outside the target locus, both eGFP and dsRED will be expressed (Figure 1B). There are also other less frequent possibilities of donor insertions that are depicted in Appendix A, but that does not affect the final readout. 

### 3.2. The Inclusion of the CRISPR/Cas9 Target Sequence into the Donor Template Increases the HDR Efficiency in K562 SEWAS84 Cells without Compromising Specificity

As a proof of principle of the suitability of the K562_SEWAS84 cell as a reporter for GE, we compare the efficacy and specificity of GE using two different donors—the SdR*GFP and the SdR*GFPWAS84—that harbour the CRISPR/Cas9 target site (WAS84) at its 5’ (Figure 2A). K562_SEWAS84 cells were nucleofected with Lg1SCas9 plasmid and each of the DNA donors (Figure 2B) and the expression of eGFP and dsRED monitored after 48 h and 30 days (or until stabilisation of eGFP and dsRED expression) (Figure 2C). As expected, the inclusion of the gRNA target site into the DNA donor increased the efficacy of HDR (around 15% with SdR*GFPWAS84 versus 1% with the SdR*GFP donor, Figure 2C,D, left plot). Interestingly, although in absolute numbers there were also higher levels of illegitimate integrations (6% versus 0.5%, Figure 2), in relative terms, the presence of the gRNA target site in the DNA donor slightly improved the specificity (right graph, Figure 2D), although in both cases over 30% of the integrations were illegitimate. One aspect to take into account is the possibility of transgene silencing as a consequence of promoter methylation, which will depend on the target cells, the promoter used and the culture conditions. It is therefore possible that this model overestimate on-target HDR GE and under-estimate illegitimate integrations since some eGFP-dsRED+ cells will be due to a combination of transgene silencing and insertion of the donor outside of the target locus. In addition, as mentioned before, the system cannot differentiate on target integrations due to HDR or by HITI. We further confirmed that eGFP+dsRED+ population was due to illegitimate integrations and eGFP-dsRED+ were on target by PCR analysis of clones derived from sorted populations (Appendix A). These analyses also showed a higher percentage of on target HITI integrations (63.6%) compared to HDR donor insertions (36.4%).

### 3.3. Generation of Cellular Models to Detect HDR-GE: eGFP-ON/dsRED-OFF Reporter

Taking into account the above-mentioned limitations of the eGFP-OFF/ds RED ON model, we aimed to generate an additional reporter cell line in which only HDR will be detected as on-target integration. HITI process should be detected as illegitimate donor integrations in order to measure precise GE by HDR. Again, this cellular model should use gRNAs that target a relevant locus (*WAS*). To generate these reporter cells, we transduced K562 cells with the GWP LVs containing in its genome a GFP-expression cassette with the first “ATG” mutated and an insertion of a fragment of the first intron from the *WAS* locus (Figure 3A, top) (see M&M for further detail). We used two different MOIs (1 and 20) of the GWP LVs and obtained bulk populations harbouring 13 vector copy number (VCN) (13-K562GWP cell line) and 0.39 VCN. Then, we isolated several clones from the 0.39 VCN bulk population to generate the 1-K562GWP reporter cell line with 1 VCN (Figure 3A). Due to the higher VCN, the 13-K562GWP reporter cell would be more sensitive, while the 1-K562GWP cell line would mimic better HDR in target cells with one copy of the target locus. Based on the integrated vectors (target locus) (Figure 3 bottom), we designed three alternative DNA donor templates (SEED, SECdR and WAS_SECdR) (Figure 3B, left), that would express only eGFP if HDR occurs (eGFP-ON/dsRED-OFF); but if inserted, outside or inside the target locus, both dsRED and eGFP will be expressed (eGFP-ON/dsRED-ON) (Figure 3B, right, and Appendix A). All these donors harbour the SFFV-eGFP expression cassette flanked by a 5’HA (5’LTR-SFFV-5’eGFP) and 3’HA (5’eGFP) and an additional dsRED expression cassette outside of the HAs. In addition, the WAS_SECdR donor contains the CRISPR/Cas9 target site (WAS intron 1) preceding the 5’HAin order to investigate its effect on the efficacy and specificity of HDR-based GE.

### 3.4. The Efficiency and Specificity of HDR-GE Is Independent on the CRISPR/Cas Delivery Method

We first use the eGFP-ON/dsRED-OFF reporter cells to investigate whether the delivery method of the Cas9/gRNA affects the efficacy and/or specificity of HDR-based GE. We nucleofected13-K562GWP cells with SEED plasmid as donor for HDR and used three different methods to deliver the CRISPR/Cas system: (1) plasmid nucleofection, (2) Cas9/gRNA ribonucleoprotein complexes (RNPs)nucleofection, and (3) transduction with all-in-one Cas9/sgRNA integration deficient LVs (IDLVs) (Figure 4A, top). eGFP and dsRED were monitored after 48 h (Figure 4A, top plots) and 30 days later (Figure 4A, bottom plots). Our data indicated that the delivery methods used to express Cas9 and gRNA into the target cell do not affect significantly the efficiency of the process (percentage of HDR-GE related to initial transfection efficacy) (Figure 4B, left graph) or specificity (Figure 4B, right graph), although a slight improvement on efficiency can be observed with RNPs nucleofection. Remarkably, in this model, over 90% of all HDR-GE were on target (Figure 4B, right graph), a specificity significantly higher than the one observed in the eGFP-OFF/dsRED-ON cellular model.

### 3.5. Analysis of GE Efficacy and Specificity of Different DNA Donor Design Using 13-K562GWP Reporter Cells

One of the crucial factors determining HDR-GE efficiency and specificity is the design of the DNA donor templates, such as the size of the homology arm or the presence/absence of the CRISPR/Cas target site. In the previous model, we had evidence that the presence of the CRISPR target site in the DNA donor increases the efficacy of on-target integrations without compromising specificity. However, the possibility of transgene silencing in that model open the possibility of an overestimation of the on-target versus the illegitimate integrations. In addition, we could not differentiate between HDR and HITI. We therefore performed a similar analysis in the 13-K562GWP reporter cells to confirm these findings. 13-K562GWP cells were nucleofected with the Lg9SCas9 plasmid expressing Cas9 and each of the DNA donors (see Figure 3B) and the expression of eGFP and dsRED were monitored after 48 h and 30 days as shown in Figure 5A. In two of the donors, dsREDcassette was under the control of CMV promoter harbouring (WAS_SECdR) or not (SECdR) the CRISPR target site. In the case of SEED donor, dsREDis expressed through the EF1alfa promoter, which is less likely to be silenced than CMV. The targeted sequence included in the WAS_SECdR donor was in reverse orientation respect to 5’HA to avoid complementarities between both regions. Our data showed that the inclusion of the CRISPR-target site into the DNA donor slightly increases the efficacy of HDR (Figure 5B, left graph) without altering or slightly improving specificity (Figure 5B, right graph). This fact indicates that the inclusion of the CRISPR-target site into the DNA donor not only increases on-target integrations by allowing donor integrations by HITI, but also enhances HDR processes. We achieved a relative HDR efficiency of 50–80% using the WAS_SECdR donor (Figure 5B, left graph) versus <5% when using SEED donor. The reason behind the low efficiency of the SEED donor is unknown. However, the slight increase in off-target integrations observed with SEED (Figure 5B, right graph) could be due to a more robust dsRED expression in this donor (due to the EF1alfa promoter) compared to SECdR-based donors (driving the expression through the CMV promoter). 

### 3.6. The Efficacy of GE Correlates with the Frequency of Target Sites

If we compared the data obtained in the K562_SEWAS84 cells with the ones obtained in the 13-K562GWP cells, we observed higher efficacy (15% versus 60%) and much better specificities (30–40% illegitimate integrations versus 0.2%) in the latest. Although this model harbours 13 target loci integrated, the K562_SEWAS84 cells only contain one target, so we wonder whether the amount of target loci per cell could be in part responsible of these findings. We therefore nucleofected the Lg9SCas9 plasmid and the WAS_SECdR donor in the 13-K562GWP and 1-K562GWP reporter cells, which harbour only one target (see Figure 3A), and compared efficacy and specificity of HDR in both cell models. Our data showed higher efficiency in the 13-K562GWP cell model (Figure 6A, left plots and Figure 6B, top graph), but similar levels of illegitimate integrations (Figure 6B, bottom graph). Therefore, the discrepancies in on-target efficacy can be explained by the increased amount of target locus in the 13-K562GWP cells compared to K562_SEWAS84 cells. However, the differences in specificity must be related to transgene silencing in the K562_SEWAS84 model, and/or to differences in DNA donor characteristics, since the 1-K562GWP model still reveals much higher specificities than the K562_SEWAS84 model. What is more, we consider that in the 1-K562GWP model the illegitimate off-target populations are underestimated because the donors that integrate through HITI into target sites light up as illegitimate integrations.

## 4. Discussion

GE strategies that involved a DNA donor should take into account not only the specificity of the nucleases, but also the fidelity of the DNA donor to carry out its function, whether it is to repair or to insert a DNA fragment into a specific place. However, current methods to quantify the fidelity of DNA donor performance are costly and/or unreliable. The main problem of the existing techniques is the low sensitivity to detect illegitimate DNA donor integrations into naturally occurring breaks or into low-abundant microhomology sites. 

In this work, we have engineered two reporter cell lines that efficiently quantify both the on-target and the illegitimate DNA donor integrations in a *WAS*-locus targeting setting. Although other reporter cells have been described to measure efficacy and specificity of GE tools [29,59,60,61], these are, to our knowledge, the first reporter systems that allow the use of gRNAs targeting an endogenous, relevant locus. This characteristic of our reporter cells allows us to compare not only efficacy and specificity of different delivery tools and different donor designs, but also different gRNAs targeted to our desired locus. However, the advantage of targeting endogenous locus in an artificial context (the integrated LV) has additional aspects to consider when interpreting the “traffic light” outcomes. Indeed, we need to consider that the endogenous locus is going to be cut by the CRISPR/Cas and, therefore the DNA donor could be integrated there by HITI. We therefore used two different approaches: a model where the on-target integrations can be due to HDR or HITI (K562_SEWAS84 cells) and another in which all the on-target integrations are due to HDR (K562GWP cells). In fact, in the case of the K562_SEWAS84 model, if HITI occurs, the cells will be eGFP-/dsRED+ just like HDR, because of the inclusion of the entire donor, interrupting eGFP expression. However, in the K562GWP cell line, if HITI occurs, the cells will be eGFP+/ dsRED+ just like illegitimate integrations and integrations in the *WAS* locus.

By using these two different models, we noticed there were strong differences in illegitimate integrations. While in the K562_SEWAS84 cells over 30% of the integrations were outside of the intended sites, in the 13-K562GWP cells, that number was below 1%in spite of the possibility, of overestimating on-target integration and underestimating illegitimate integrations. Since our initial experiments were performed in K562_SEWAS84 cells (with 1VCN) and 13-K562GWP cells (with 13VCN), we initially reasoned that the amount of target loci per cell could be, in part, responsible of the enhanced specificity of the latest model. However, we found similar specificities in 1-K562GWP cells harbouring one single target per cell. Earlier studies targeting the endogenous *WAS* locus using the same sgRNA [28] indicated higher out-target integrations, closer to the amounts found in the K562_SEWAS84 model. We suspect that the differences found in specificity between earliest findings [28], the K562_SEWAS84 model and the K562GWP cells can be related to factors such as dsRED transgene silencing in the donor used in this model and/or to differences in DNA donor characteristics.

Our models allowed us to determine that the inclusion of the CRISPR/Cas9 target in the DNA donor substantially improved the efficiency of on-target integration without negatively affecting the specificity. Particularly interesting was the finding that it can improved efficacy of HDR, not only HITI, thanks to the K562GWP cellular model. These results match with the approaches proposed by other groups, where the inclusion of the target sequence in the DNA donor increases HDR efficiency by two-fold to fivefold regarding circular plasmid [45,47], as well as the integration of the DNA donor by HITI [64,65]. We must consider that our models can only be used with large donors containing a reporter expressing cassette. Therefore, we cannot study the specificity of genome editing strategies using ssODN in our models.

Finally, our K562GWP cells harbouring 13 (13-K562GWP cells) or 1 (1-K562GWP cells) target loci per cell allowed us to investigate the effect of the frequency of the target locus on GE. We observed an increase in efficacy without significantly affecting specificity. This result confirms what it has been said in other articles about the cell response to CRISPR-Cas9 editing strongly correlated with the number of target loci [66]. In our work, we have used the *WAS* gene as a relevant sequence in our reporter cellular models, however, our models could change the *WAS* target sequence to another gene of interest. These cellular models would allow to further investigate new strategies to improve HDR or HITI efficacies and specificities for targeting any gene. 

To summarise, we have generated two reporter cellular models to study the efficiency and specificity of genome editing strategies using large donor templates. By using these two models and a fluorescence-based pattern we have shown that the HDR specificity is independent of the delivery method and that the inclusion of the target site in the DNA donor enhances efficiency without affecting specificity. Finally, we also showed that the copy number of the target site influences specificity and efficacy of genome editing.

## Figures and Tables

**Figure 1 cells-09-01492-f001:**
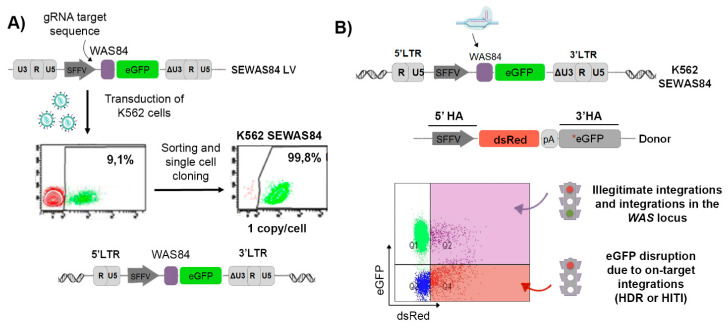
Development of K562 SEWAS84 reporter cell line (eGFP-OFF/dsRED-ON). (**A**) Generation of the K562 SEWAS84 reporter cell line. Top: Representative scheme of the SEWAS84 lentiviral plasmid. Left: Representative plots of K562 cells transduced with SEWAS84 LVs at MOI=0.1. The eGFP+ population was sorted to generate the K562 SEWAS84 cell line expressing eGFP and containing 1 copy was obtained by single-cell cloning. (**B**) Scheme of the strategy for targeting the WAS sequences to silence eGFP in order to have an easy readout of the efficacy and specificity in the K562SEWAS84 cellular model. The sgRNA will cut in the WAS84 region of the integrated LV (top) promoting the on-target integration in the presence of appropriate DNA donors (bottom). The design of the DNA donor, harbouring an expression cassette for dsRED and a mutated eGFP flanked by the homology arms (5’HA and 3’HA), allow to measure efficiency of on-target donor integrations (% of GFP- dsRED+), as well as frequency of illegitimate insertion (% of eGFP+ dsRED+) as illustrated at the bottom of the figure. Nucleofection of plasmid in the absence of the CRISPR/Cas was used as control for background (se M&M for details).

**Figure 2 cells-09-01492-f002:**
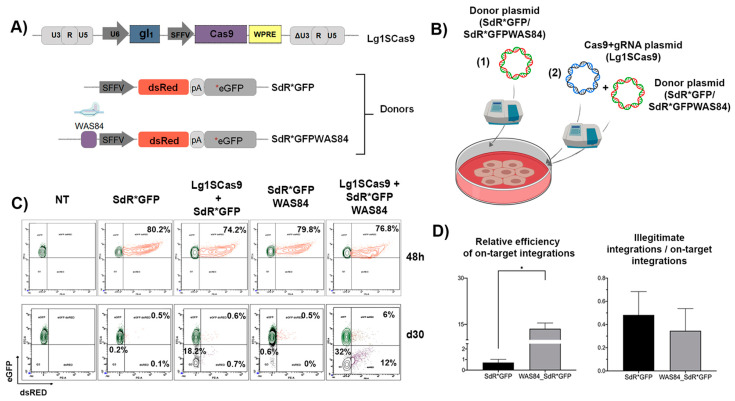
Inclusion of the target sequence into the donor template increases the efficiency of on-target donor integration without increasing illegitimate integrations in K562 SEWAS84. (**A**) Scheme of CRISPR/Cas9 lentiviral plasmid expressing Cas9 and the gRNA (See M&M for details) (top) and different DNA donors (bottom): SdR*GFP donor (previously describe) and SdR*GFPWAS84 donor harbouring the CRISPR/Cas9 target site. (**B**) Drawing of the delivery methods used for the different donors (SdR*GFP donor or SdR*GFPWAS84) and the CRISPR/Cas9 system (Lg1SCas9) used alone as controls (1) and with the CRISPR/Cas9 system (Lg1SCas9) (2). Both systems were delivered to the target cells as plasmids by nucleofection. (**C**) Representative plots showing eGFP (FITC-A) and dsRED (PE-A) expression 48 h (top) and 30 days (bottom) of K562 SEWAS84 without any treatment (NT) and after nucleofection with donors only (SdR*GFP donor or SdR*GFPWAS84), and with L1SCas9 (all-in-one CRISPR/Cas9 lentiviral plasmid expressing Cas9 and the gRNA. See M&M for further detail) and SdR*GFP or SdR*GFPWAS84 donor as indicated on the top of each graph. with all-in-one CRISPR/Cas9 lentiviral plasmid expressing Cas9 and the xgRNA (See M&M for further detail) and SdR*GFP donor (left) or SdR*GFPWAS84 donor, (right). (**D**) Graphs showing the relative GE efficacy and specificity for both DNA donors. The relative efficacy of GE (left graph) was calculated by dividing the % of eGFP-dsRED+ cells (due to HDR or HITI) at day 30 by the % of double positive cells at 48 h (initial transfection efficacy). Right graph shows the frequency of illegitimate integration related to the efficacy of HDR. Values represent mean of three separate experiments and the error bars indicates the standard errors of the mean. The comparison between the different groups was carried out using the Mann–Whitney U test (* *p* < 0.05).

**Figure 3 cells-09-01492-f003:**
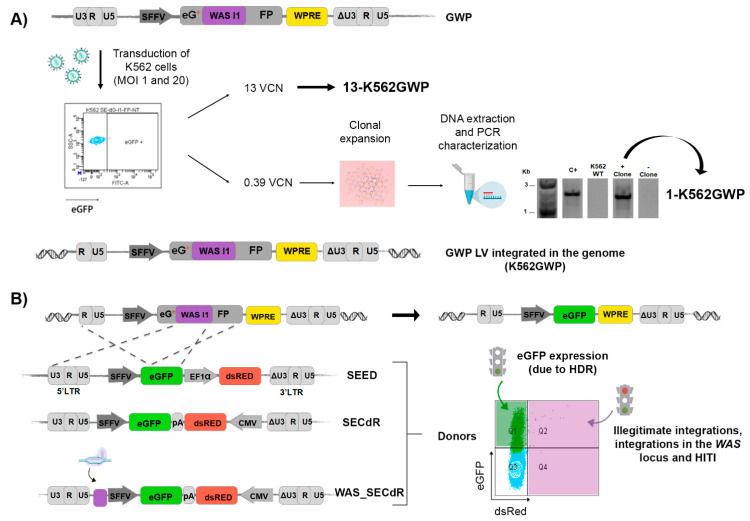
Development of K562GWP reporter cell lines (eGFP-ON/dsRED-OFF). (**A**) Scheme showing the procedure to generation of the K562GWP reporter cell lines. Top: Representative scheme of the GWP lentiviral plasmid used to produced LVs to transduce K562 cells (bottom). We used two different MOIs (1 and20) of the GWP LVs and obtained bulk populations harbouring 13VCN (13-K562GWP cell line) and 0.39 VCN. A representative plot of K562 cells transduced with GWP LV is shown to indicate absence of eGFP expression. The 0.39 VCN population was cloned and characterised in order to obtain 1 VCN (1-K562GWP reporter cell line). A scheme of the integrated vector (the target locus) is shown at the bottom. (**B**) Scheme of the strategy for targeting the *WAS* sequence to rescue eGFP expression through HDR using three different DNA donors; SEED, SECdR and WAS_SECdR. The sgRNA will cut in the *WASI1* region (in purple) of the integrated LV (top) promoting the HDR process and giving rise to GFP+dsRED- cells (top-right) due to the presence of the SFFV-eGFP expression cassette flanked by a 5’HA and 3’HA. All donors have an additional dsRED expression cassette outside of the HAs which allow to measure specificity of HDR, asdonor insertions through a non-HDR mechanism will render GFP+ dsRED+ cells (expressed by the DNA donors), plot at the right.

**Figure 4 cells-09-01492-f004:**
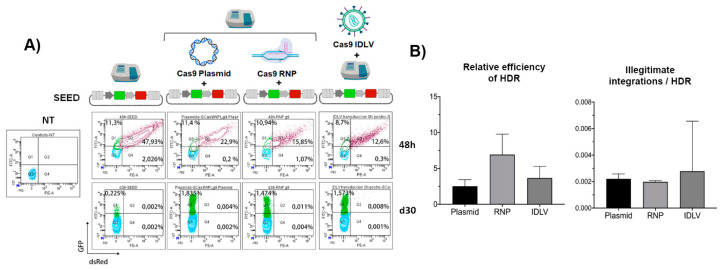
CRISPR/Cas9 delivery systems achieved similar efficacies and specificities of HDR in K562GWP model. (**A**) Representative cytometry plots showing the differential fluorescence patterns of edited K562GWP cells, using different methods to deliver CRISPR/Cas9 system. The top of the figure illustrates the different delivery methods used to transfer CRISPR/Cas9 system to K562GWP cells: plasmid nucleofection, Cas9/gRNA ribonucleoprotein complexes (RNPs) nucleofection, and transduction with all-in-one Cas9/sgRNA IDLVs. The SEED plasmid donor was delivered by nucleofection, alone (as a negative control, left plots) or in combination with the CRISPR/Cas9. eGFP and dsRED levels were analysed at 48 h (upper plots) and at day 30 (lower plots). (**B**) Graphs showing the relative HDR efficacy (left) and specificity (right) of HDR for different delivery methods. The relative efficacy of HDR (left graph) was calculated by dividing the % of eGFP+dsRED- cells (on-target HDR) at day 30 by the % of double positive cells at 48 h (initial transfection efficacy). Right graph shows the frequency of illegitimate integration related to the efficacy of HDR. Values represent mean of three separate experiments and the error bars indicates the standard errors of the mean. The comparison between the different groups was carried out using the Mann–Whitney U test (* *p* < 0.05).

**Figure 5 cells-09-01492-f005:**
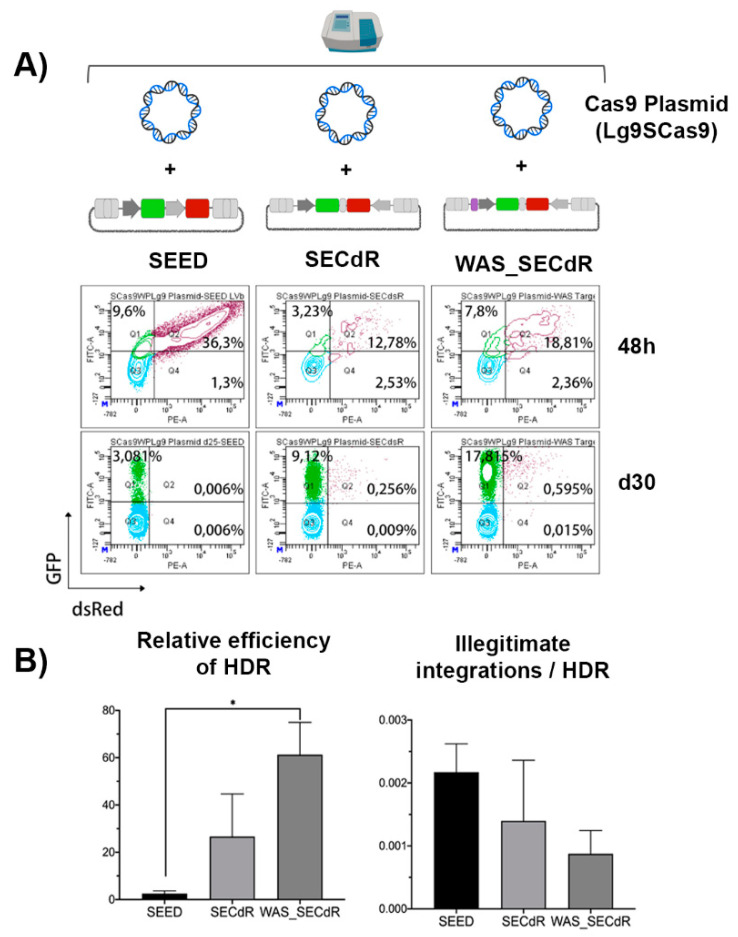
Efficacy and specificity of HDR in K562GWP cell line with different donor templates. (**A**) Representative cytometry plots showing the expression of eGFP (FITC-A) and dsRED (PE-A) in 13-K562GWP cells over time. Each of the donor templates (SEED, SECdR or WAS_SECdR), as well as the CRISPR/Cas9 system, was transferred to the target cells as plasmids by co-nucleofection. The expression of eGFP and dsRED was determined 48 h post-nucleofection, and 30 days later. (**B**) Graphs showing the relative HDR efficacy and specificity of HDR-GE for the different DNA donors. The relative efficacy of HDR (left graph) was calculated by dividing the % of eGFP+dsRED- cells (on-target HDR) at day 30 by the % of double positive cells at 48 h (initial transfection efficacy). Right graph shows the frequency of illegitimate integration related to the efficacy of HDR. Values represent mean of three separate experiments and the error bars indicates the standard errors of the mean. The comparison between the different groups was carried out using the Mann-Whitney U test (* *p* < 0.05).

**Figure 6 cells-09-01492-f006:**
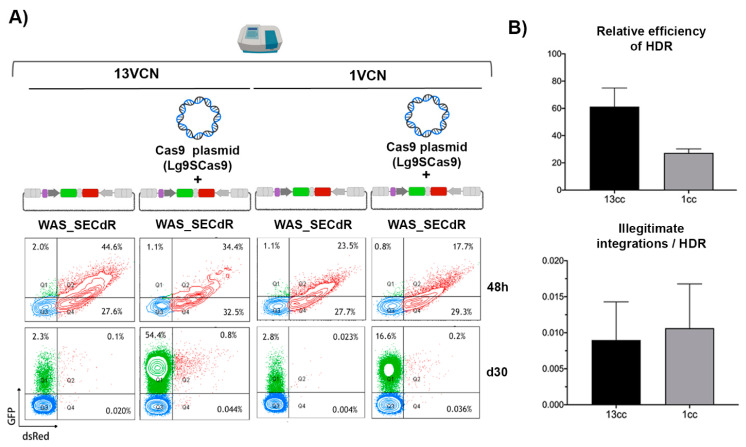
Correlation of the efficacy of GE with the frequency of target sites. (**A**) Representative cytometry plots showing the expression of eGFP and dsREDon 13-K562GWP cells (13VCN) and 1-K562GWP (1VCN) over time. Reporter cells were nucleofected with WAS_SECdR donor alone or together with the Lg9SCas9 plasmid as indicated in the figure and the percentage of eGFP and dsRED expressing cells were determined 48 h (top plots) and 30 days (bottom plots) post-nucleofection. (**B**) Graphs showing the relative HDR efficacy (top) and specificity (bottom) of HDR for 13-K562GWP cells (13cc) and 1-K562GWP (1cc) reporter cells. The relative efficacy of HDR was calculated by dividing the % of eGFP+dsRED- cells (on-target HDR) at day 30 by the % of double positive cells at 48 h (initial transfection efficacy). The frequency of illegitimate integration is shown related to the efficacy of HDR. Values represent the mean of three separate experiments and the error bars indicate the standard errors of the mean. The comparison between the different groups was carried out using the Mann–Whitney U test (* *p* < 0.05).

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
