# Peer review of "Development of Cellular Models to Study Efficiency and Safety of Gene Edition by Homologous Directed Recombination Using the CRISPR/Cas9 System"

_cells, 2020, doi:10.3390/cells9061492_

Round 1
Reviewer 1 Report
The authors engineered two cellular models, and used these to show that the specificity of HDR-mediated CRISPR/Cas9 mutations are independent of delivery method; that the insertion of the target sequence into the donor DNA affect (enhance) efficiency but not specificity; and that copy number of the target site influence specificity and efficacy of CRISPR/Cas9-mediated genome-editing.
This study provides relevant data on safety issues related to knock-in mutations rather than knock-out mutations. Given that the available studies (and they are really few) in this area have focused mainly on knock-out mutations. Thus, this is a timely study, which provides important resources for investigations into the safety and impact of CRISPR/Cas9.
Few minor punctuation and spell checks are required:
Line 84 - Punctuation (.)
Line 82 -ímproved´
Line 71: Define ´SDN`
Line 91: `Donor DNAs` OR `Donor´s DNA`
Author Response
Reviewer 1. Comment 1
The authors engineered two cellular models, and used these to show that the specificity of HDR-mediated CRISPR/Cas9 mutations are independent of delivery method; that the insertion of the target sequence into the donor DNA affect (enhance) efficiency but not specificity; and that copy number of the target site influence specificity and efficacy of CRISPR/Cas9-mediated genome-editing. This study provides relevant data on safety issues related to knock-in mutations rather than knock-out mutations. Given that the available studies (and they are really few) in this area have focused mainly on knock-out mutations. Thus, this is a timely study, which provides important resources for investigations into the safety and impact of CRISPR/Cas9.
Author answer: We thanks the reviewer for the nice comments
Reviewer 1. Comment 2
Few minor punctuation and spell checks are required:
Line 84 - Punctuation (.)
Author answer: Done. Now in line 76
Line 82
Author answer: Changed in line 81 of the revised manuscript
Line 71: Define ´SDN`.
Author answer: ssODNs has now been defined in line 79 of the revised manuscript
Line 91: `Donor DNAs` OR `Donor´s DNA`
Author answer: We have included “DNA donor” all through the text.
Reviewer 2 Report
The authors of this manuscript describe fluorescence reporter models to detect on-target and off-target editing by CRISPR-HDR based gene editing. The system is new and interesting, however, it fails to convince the importance and broader interest of this system to the audience. First of all, off-target editing mainly comes from gRNA-mediated genome cutting, so interests in the field are focused on the detection of that step. More importantly, the clinical interest for CRISPR-based technology currently focuses on gene mutation or correction within a short sequence, instead of the inclusion of a large gene fragment. The reporter system this manuscript uses relies on lengthy FP, which can be a major bias for HDR as the longer sequence insertion leads to lower possibility of integration. The author would also need to compare with the upfront technologies for CRISPR specificity detection, for example, GUIDE-seq or CIRCLE-seq.
In terms of writing, the abstract and introduction is not organized in an efficient logic to highlight the importance and broader interest that findings in this manuscript can bring to the audience. In several occasions, the writing is hard to follow or presents incorrect statements, e.g. line 17-19, 23-24, 40-42, 53-54, 69, 83-85, 110-112. The authors should rewrite the introduction so that claims and related scientific background are sufficient and closely related with the findings in this work.
A major problem of the data is that it lacks a control of donor plasmid only transfection, because the donor has SFFV promoter and can be expressed the dsRed gene (Figure 1B) without integration into the genome. Also the author needs to confirm that the GFP+/dsRed+ cell population are non-specifically integrated cells.
Author Response
Reviewer 2. Comment 1
The authors of this manuscript describe fluorescence reporter models to detect on-target and off-target editing by CRISPR-HDR based gene editing. The system is new and interesting, however, it fails to convince the importance and broader interest of this system to the audience. First of all, off-target editing mainly comes from gRNA-mediated genome cutting, so interests in the field are focused on the detection of that step.
Author answer. We do agree that off-target cuts are one of the main problems of genome editing technologies and the main focus in the field. However, several promising GE strategies use donor DNAs and it is fundamental to have a clear vision of their efficacy and safety.
Reviewer 2. Comment 2
More importantly, the clinical interest for CRISPR-based technology currently focuses on gene mutation or correction within a short sequence, instead of the inclusion of a large gene fragment. The reporter system this manuscript uses relies on lengthy FP, which can be a major bias for HDR as the longer sequence insertion leads to lower possibility of integration. The author would also need to compare with the upfront technologies for CRISPR specificity detection, for example, GUIDE-seq or CIRCLE-seq.
Author answer. Indeed, longer sequences leads to lower possibility of integration, but at the same time, highest specificities. We have included new text in discussion to cover this point (lines 515-517). However, although it would be very interesting to compare our data with GUIDE-seq or CIRCLE-seq, these experiments are time consuming (we only have 10 days for answering) and also costly. However we will take into consideration these technologies for future experiments.
Reviewer 2. Comment 3
In terms of writing, the abstract and introduction is not organized in an efficient logic to highlight the importance and broader interest that findings in this manuscript can bring to the audience. In several occasions, the writing is hard to follow or presents incorrect statements, e.g. line 17-19, 23-24, 40-42, 53-54, 69, 83-85, 110-112. The authors should rewrite the introduction so that claims and related scientific background are sufficient and closely related with the findings in this work.
Author answer. We thanks the reviewer for these comments. Thw mauscript have now been revised by a native-speaking inglish and we have re-write the abstract (lines 20-34) and introduction (lines 58-62; lines 77-80 and lines 118-121) to make it clearer to the reader, highlighting the importance of the findings. We have also correct the different sentences that the reviewer found misleading:
lines 17-19 have been modified and are now lines 18-20 in the revised manuscript.
lines 40-42 have been modified and are now lines 44-45 in the revised manuscript.
lines 53-54 have been modified and are now lines 57-62 in the revised manuscript.
lines 69 have been modified and are now lines 76-79 in the revised manuscript.
lines 83-85 have been modified and are now lines 90-94 in the revised manuscript.
lines 110-112 have been modified and are now lines 116-119 in the revised manuscript.
Reviewer 2. Comment 4
A major problem of the data is that it lacks a control of donor plasmid only transfection, because the donor has SFFV promoter and can be expressed the dsRed gene (Figure 1B) without integration into the genome.
Author answer. We are sorry about the misunderstanding. All experiments were performed using Donor-only controls. We have modified legend of Figure 1 (lines 292-293) and Figure 2 (Lines 324-325) and made a new Figure 2B to make it clearer. We have also modified the material and methods (lines 240-242 and lines 248-262) to make it clearer to the reader.
Reviewer 2. Comment 5
Also the author needs to confirm that the GFP+/dsRed+ cell population are non-specifically integrated cells.
Author answer. We Thanks the reviewer for the comment. We did previously sorting and cloning of GFP+dsRed+ and GFP-dsRed+ to analyze insertion sites of the different populations. We have recently finished the analysis and have now included these data. We have re-write the results (lines 314-317) and included a new supplementary figure (Figure S2). PCR analysis of the different clonal populations showed that the eGFP+dsRED+ population was due to illegitimate integrations while eGFP-dsRED+ cells were on-target integrations due to HITI (63.6%) or HDR (36.4%).
Round 2
Reviewer 2 Report
Thank you for the responses to my comments, but I still preserve my attitude toward the following two points.
(1) A major part of clinical interest for CRISPR-based technology focuses on gene mutation or correction within a short sequence, instead of the inclusion of a large gene fragment. The reporter system this manuscript uses relies on lengthy FP, which can be a major bias for HDR as the longer sequence insertion leads to lower possibility of integration. The enhanced specificity as the author responded does not matter as this platform is designed as a method to detect off-target insertion effect. The authors need to refine their introduction to limit the scope of the manuscript on lengthy gene insertion instead of short DNA template based HDR, as this system can not reflect off-target insertion of short templates.
(2) In terms of donor only controls discussed in comment 4, it is not enough to just include a general description. The author must provide data showing two important negative controls: 1, Cell only without any treatment 2, Cell transfected with donor plasmids but not Cas9. These negative controls need to be included in the main figure.
For the newly added supporting figures, they are not available to me. So I could not comment on those.
Author Response
Reviewer comment (1). A major part of clinical interest for CRISPR-based technology focuses on gene mutation or correction within a short sequence, instead of the inclusion of a large gene fragment. The reporter system this manuscript uses relies on lengthy FP, which can be a major bias for HDR as the longer sequence insertion leads to lower possibility of integration. The enhanced specificity as the author responded does not matter as this platform is designed as a method to detect off-target insertion effect. The authors need to refine their introduction to limit the scope of the manuscript on lengthy gene insertion instead of short DNA template based HDR, as this system can not reflect off-target insertion of short templates.
Authors´answer: Thank you for the comment. We have modified the text in introduction (lines 75-82, and lines 122, 125 and 128) in the revised manuscript to limit the scope to long DNA donors as suggested by the reviewer.
Reviewer comment (2). In terms of donor only controls discussed in comment 4, it is not enough to just include a general description. The author must provide data showing two important negative controls: 1, Cell only without any treatment 2, Cell transfected with donor plasmids but not Cas9. These negative controls need to be included in the main figure.
Authors´answer: Thank you for the comment. We have modified Figure 2C in the revised version to include the controls suggested by the reviewer: NT = cell only without any treatment; SdR*GFP and SdR*GFPWAS84 = Cells transfected with donor plasmid but not Cas9. We have also modified the text in figure 2 legend and in line 355 to further clarify this point.